# Unlocking Coherent Reasoning in LLMs with Hierarchical Soft Prompts

## Abstract

Large language models (LLMs) exhibit strong reasoning capabilities in complex tasks. Soft prompt tuning, as a lightweight approach, injects trainable vectors into the input to guide the reasoning process and enhance model performance. Prior studies show that soft prompts effectively activate prior knowledge and improve problem understanding in the early stages of reasoning. However, when they continue to exert strong influence in the middle and later stages, they often disrupt the information flow and degrade reasoning performance. Based on this observation, we argue that the role of soft prompts should not be confined to a single stage of activation and guidance. Instead, they should be inserted at appropriate stages to ensure smooth information transmission across layers. Existing methods, however, typically rely on one-shot static injection and cannot dynamically regulate prompts across stages, leading to functional mismatches during reasoning. To address this limitation, we propose a dynamic hierarchy-aware mechanism(DHAM). This mechanism first employs hierarchical clustering to derive stage-specific representations, and then leverages the semantic guidance capability of soft prompts to adaptively align and activate them, ensuring effective coordination across reasoning stages. DHAM yields consistent gains across models and benchmarks (e.g., 29.5%→43.8% on Llama-2-13B/GSM8K), with ablations showing CKA clustering and moderate stage numbers (e.g., $G = 3/4$) perform best, consistent with the stable information flow hypothesis.

## 1 Introduction

Large Language Models (LLMs) (Vaswani et al., 2017; Brown et al., 2020; Raffel et al., 2020) demonstrate strong capabilities in knowledge integration and reasoning across open-domain question answering, mathematical reasoning, and multi-hop inference tasks. However, improving their reasoning performance without incurring large parameter overhead remains challenging. As a parameter-efficient paradigm, Soft Prompting has received increasing attention due to its lightweight nature, transferability, and training efficiency (Li & Liang, 2021). By injecting learnable prompt vectors into the input, this approach allows models to rapidly adapt to downstream tasks while keeping the backbone parameters frozen.

Nevertheless, existing soft prompt methods predominantly adopt a static injection strategy, where prompt vectors are introduced into model layers in a heuristic or intuition-driven manner rather than being dynamically adapted (Lester et al., 2021; Liu et al., 2021; 2023). Although such designs can activate prior knowledge and facilitate problem understanding, their persistent and non-adaptive influence often leads to over-reliance on the prompts themselves, thereby disrupting information flow and weakening logical integration (Dai et al., 2021; Wang et al., 2023; Yuan et al., 2024). In other words, static prompting cannot dynamically adapt to the reasoning process, and in complex reasoning tasks, it frequently causes late-stage mismatches, reducing both coherence and stability.

To address this issue, researchers have explored multiple improvement directions. Some studies extend continuous prompts across multiple layers to approximate full fine-tuning (Liu et al., 2021), while others introduce late prompts at intermediate layers to strengthen information flow control in later reasoning (Liu et al., 2022). Additional approaches learn where to place prompts and how strongly they should act across layers (Zhu & Tan, 2023), or dynamically determine the length, position, and representation of prompts from an instance-specific detailed perspective (Wu et al.,

2022; Yang et al., 2023). Recently, several works have proposed detecting and masking harmful prompts during reasoning to mitigate negative effects (Fan et al., 2025). Although these methods make progress in layer selection, gating, and instance adaptation, they generally focus on single-point optimization or instance-level adjustment and still lack explicit stage modeling of the reasoning process as well as mechanisms for aligning information flow.

We argue that improving complex reasoning performance requires not only designing better prompts but also capturing the hierarchical structure of the reasoning process and aligning prompts with stage-level information. In other words, an explicit stage-aware scheduling mechanism is needed to dynamically adapt to reasoning requirements at different stages. To this end, we propose the Dynamic Hierarchy-Aware Mechanism (DHAM).

Specifically, we first use Centered Kernel Alignment (CKA) similarity to measure relationships between different layer representations, and then apply hierarchical clustering to partition the multi-layer hidden states of the model into several stage-wise groups, each capturing functionally similar layers during reasoning. We then introduce trainable soft prompts into each stage and jointly train them with the corresponding stage representations, ensuring stable information transmission and dynamic alignment within stages. Compared with static full-layer injection, single late insertion, gated layer selection, instance-level adaptation, and harmful prompt masking, our method builds on significance-driven process diagnostics to achieve fine-grained prompt injection at the hierarchical level. This design effectively alleviates late-stage mismatches, improves the coherence and stability of reasoning flows, and ultimately enhances reasoning accuracy in complex tasks. In summary, our main contributions are as follows:

- We conduct saliency-score-based diagnostics and reveal stage-wise trends in information flow, further identifying patterns that are beneficial for reasoning.
- Based on this finding, we propose a CKA-driven hierarchical clustering method together with a stage-level soft prompt scheduling mechanism, which dynamically aligns information flow and injects prompts within the hierarchical structure, effectively mitigating the mismatch problem in static prompting methods.
- We design and carry out comprehensive experimental evaluations on multiple complex reasoning tasks, and the results demonstrate that our approach consistently outperforms existing prompt-tuning methods in terms of reasoning coherence and accuracy.

## 2 RELATED WORK

**Prompt-based Adaptation for LLM Reasoning.** In recent years, researchers have widely adopted Prompt Tuning, especially Soft Prompt tuning, as a parameter-efficient adaptation method (Liu et al., 2021; Lester et al., 2021; Ding et al., 2023). Studies show that Prompt Tuning enhances downstream task performance with only a small number of trainable parameters while keeping the pretrained backbone frozen (Li & Liang, 2021; Liu et al., 2023). However, the role of Soft Prompts across different reasoning stages remains unclear, which makes it challenging to leverage them effectively in complex reasoning.

Researchers have therefore developed dynamic control mechanisms to improve the adaptability of soft prompts. Instance-adaptive Prompting (Yuan et al., 2024) selects prompts for each input instance, and the Dynamic Prompting framework (Yang et al., 2023) explores dynamic positions, lengths, and prompt pools. Other methods, such as Adaptive Prefix Tuning (APT) (Zhang et al., 2023) and Hierarchical Prompt Tuning (HPT) (Wang et al., 2022; Zeng et al., 2024), incorporate hierarchical information or gating mechanisms to differentiate the effects of prompts across layers and semantic stages. More recently, Fan et al. proposed Dynamic Prompt Corruption (DPC), which uses saliency analysis to detect harmful prompts in later reasoning stages and applies dynamic masking to mitigate their effects (Fan et al., 2025).

**Information-Flow Analyses and Stage-Aware Motivation.** To investigate how soft prompts influence reasoning, researchers have applied information flow and saliency analyses (Simonyan et al., 2013; Selvaraju et al., 2017; Abnar & Zuidema, 2020). Dai et al. introduced Knowledge Neurons to characterize knowledge storage units inside LLMs (Dai et al., 2021), and Wang et al. analyzed in-context learning from an information flow perspective (Wang et al., 2023). Beyond raw attention maps, works have quantified how information propagates through Transformer layers (e.g.,

attention rollout/flow and cross-layer relevance propagation) (Abnar & Zuidema, 2020; Chefer et al., 2021). Complementary studies analyzed what attention heads "look at" and whether attention is an explanation, revealing specialized heads and mixed evidence on attention's explanatory power (Clark et al., 2019; Voita et al., 2019; Jain & Wallace, 2019; Wiegreffe & Pinter, 2019). Drawing on evidence across prior studies, we observe a consistent tendency: across a range of settings, successful reasoning shifts saliency from prompts to the question and intermediate steps, whereas unsuccessful reasoning exhibits stronger prompt dependence in deeper layers, thereby disrupting the coherence of information flow. This observation motivates us to differentiate the role of soft prompts across hierarchical levels. Accordingly, we propose DHAM and describe it below.

## 3 PRELIMINARY

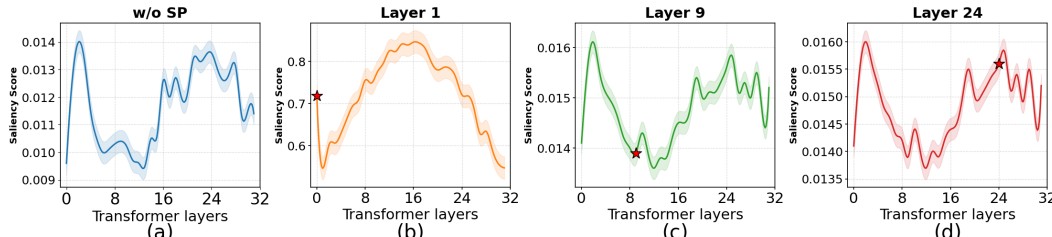

Figure 1: Layer-wise saliency with and without soft prompt insertion. The pretrained model (a, w/o SP) shows oscillatory peaks. Early SP at Layer 1 (b) yields a smooth rise–fall trajectory, while mid/late SP at Layers 9 and 24 (c, d) introduce spikes and backflows, indicating less stable transmission and motivating stage-aware prompt scheduling.

Prior studies show that soft prompt may play different roles at different stages of reasoning, and that they may introduce interference effects in the middle and late stages. While existing work reveals this phenomenon through empirical results, the underlying mechanisms remain insufficiently understood. To systematically understand the true impact of soft prompts on the reasoning process of large models, we examine the information flow across Transformer layers. Specifically, this section aims to answer two key questions:

(1) *How does information propagate and evolve with depth when soft prompts are inserted at different layers?*

(2) *Which inter-layer transmission pattern better maintains stable information flow and improves complex reasoning performance?*

To this end, we construct both a visualization and a quantitative analysis of inter-layer information flow based on saliency scores Dai et al. (2021). Figure 1 illustrates the visualization (more visualizations of saliency-based information flow can be found in the Appendix A.2), where saliency is defined as:

$$I^l = \sum_h A^{h,l} \odot \frac{\partial L(x)}{\partial A^{h,l}} \tag{1}$$

where $A^{h,l}$ denotes the attention matrix of the $h$-th head in the $l$-th layer, $L(x)$ denotes the task loss (cross-entropy), and $\odot$ represents element-wise multiplication. For visualization, we aggregate and normalize across heads and positions (taking the absolute value and averaging), obtaining a single scalar for each layer and plotting its variation with respect to layer depth (red markers indicate insertion layers). To eliminate the confounding factor of correctness and focus on the shape of information flow trajectories, we only visualize samples that are correctly solved under all four configurations. This allows us to directly observe the influence of different insertion stages without interference from task difficulty or outcome differences.

Under this controlled setting, as shown in Figure 1, we observe significant differences in saliency curve shapes across different insertion stages. (a) The "w" configuration shows a relatively unstable and erratic pattern, indicating a less stable inter-layer information flow. (b) Early insertion of soft prompts results in a unimodal and smooth trajectory that rises and then falls, suggesting

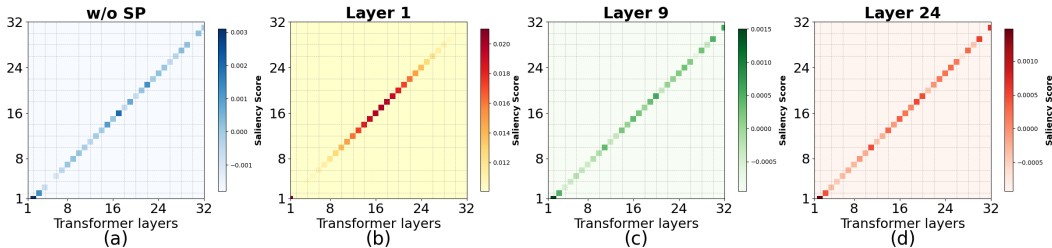

Figure 2: Layer-wise saliency difference heatmaps under different SP insertion settings, computed as the change between consecutive layers (later minus earlier). The pretrained model (a, w/o SP) and mid/late insertions (c, d) show scattered hotspots and irregular perturbations, indicating unstable propagation. Early insertion at Layer 1 (b) yields a smooth diagonal band, suggesting stable cross-layer information flow.

a gradual migration of information from the prompt to the problem statement and intermediate reasoning steps as depth increases, thereby forming a more stable flow. (c) Mid-stage insertion leads to multi-peaked and oscillatory patterns with frequent spikes and local reversals, signifying unstable information transmission across layers. (d) Similarly, late-stage insertion also exhibits a highly oscillatory pattern, with significant fluctuations and reversals in the saliency curve, indicating stage misalignment and disrupted flow of information. These observations indicate that the insertion stage systematically alters the way information propagates across layers.

We further quantify these transmission patterns by visualizing saliency differences between adjacent layers (Figure 2). We find that early insertion significantly reshapes the inter-layer structure: the diagonal band becomes more continuous and uniform, and scattered hotspots are reduced, indicating that saliency is smoothly propagated between adjacent layers and that abrupt cross-stage shifts (i.e., backflows) are suppressed. By contrast, the other three configurations show highly similar structures, consistent with their oscillatory saliency trajectories. Combined with the accuracy comparison on GSM8K using LLaMA-3-8B (Figure 3), where early insertion achieves the highest accuracy of 70.7%, we conclude that smooth migration patterns are likely more beneficial for reasoning performance.

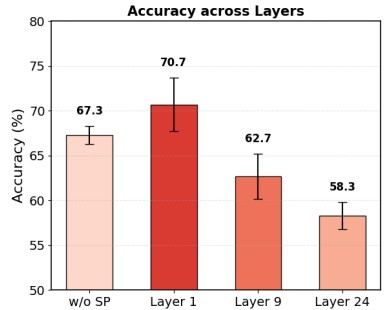

Figure 3: Accuracy comparison under different SP insertion positions on GSM8K with LLaMA-3-8B.

To formalize this relationship, we propose two testable hypotheses: (i) the Stable Information Flow Hypothesis—a unimodal and smooth migration of saliency along depth facilitates controllable reasoning paths and stronger robustness; and (ii) the Stage Misalignment Hypothesis—when soft prompts continuously dominate saliency in the middle and late stages, inducing repeated backflows and spikes, model attention competes with task signals at inappropriate stages. This disrupts information propagation, occasionally yielding correct answers but with weaker robustness and generalization.

If these hypotheses hold, we expect to observe consistent evidence across datasets and models: the smoother and more unimodal the saliency curve, the more robust the sample is to paraphrasing or mild noise; moving soft prompts from later to earlier stages, or suppressing their influence in later stages, shifts the curve from a multi-peaked oscillatory pattern to a unimodal smooth pattern; and other interpretability signals (e.g., attention rollout, cross-layer correlation propagation) more consistently trace to the problem statement and intermediate reasoning steps, rather than repeatedly returning to the prompt itself.

In summary, even when the final answers are identical, an along-depth smooth migration pattern aligns better with the information flow required for coherent reasoning, whereas repeated backflows and spikes reflect stage misalignment and attention competition. This observation directly motivates the method design in the following section: by differentiating the roles and strengths of soft prompts

across stages and suppressing their excessive influence in the later layers, we promote stable inter-layer propagation and thereby improve reasoning accuracy.

# 4 METHOD

We propose a DHAM that models cross-layer organization in LLMs by clustering layers into coherent stages and injecting trainable soft prompts at representative layers to provide stage-specific semantic guidance during inference. In the following sections, we detail the overall workflow and the key technical elements of DHAM.

## 4.1 HIERARCHICAL PARTITIONING

A large language model typically contains $L$ Transformer layers with hidden dimension $d$. For an input sequence of length $n$, the output of the $l$-th layer is denoted as $X^{(l)} \in \mathbb{R}^{n \times d}$. To measure the similarity among internal representations, we adopt Centered Kernel Alignment (CKA) (Kornblith et al., 2019), a normalized dependence measure widely used for comparing neural network features. We choose CKA as it is more effective in capturing cross-layer distributional similarity and more stable for hierarchical partitioning than cosine similarity or other common measures. Given two representations $X, Y \in \mathbb{R}^{n \times d}$, the CKA score is defined as:

$$\text{CKA}(K, L) = \frac{\text{HSIC}(K, L)}{\sqrt{\text{HSIC}(K, K)\, \text{HSIC}(L, L)}} \tag{2}$$

where $K = XX^\top$, $L = YY^\top$, and HSIC denotes the Hilbert–Schmidt Independence Criterion.

After obtaining the similarity matrix, we compute pairwise CKA scores across the $L$ layers to construct $S \in [0, 1]^{L \times L}$. We then apply agglomerative hierarchical clustering (Murtagh & Contreras, 2012) to partition the model into $G$ hierarchies $\{\mathcal{G}_1, \ldots, \mathcal{G}_G\}$ (see Fig. 4). Each hierarchy corresponds to a group of layers with similar functional roles in reasoning and is regarded as a semantic unit.

The number of hierarchies $G$ is determined in a data-driven manner. Specifically, we employ hierarchical clustering on the CKA-based similarity matrix to construct a dendrogram $\mathcal{T}$, which models the aggregation relations among layers. This method does not require a pre-specified number of clusters, and the resulting tree structure inherently provides multi-granularity hierarchical partitions. Such partitions offer interpretable layer groupings at different depths, thereby supplying a stable structural basis for subsequent stage-wise soft prompt injection.

On the dendrogram $\mathcal{T}$, a cut threshold $\tau$ produces the initial number of clusters as:

$$G(\tau) = \text{NumClusters}(\mathcal{T}, \tau) \tag{3}$$

corresponding to different hierarchical partitions. To select the optimal hierarchy number, we compute the Silhouette coefficient for each partition and take the best-performing one as:

$$G^\star = \arg\max_{G(\tau)} \text{Silhouette}(G(\tau)) \tag{4}$$

Furthermore, to mitigate the instability caused by sample distribution randomness, we adopt bootstrap resampling and choose the value of $G$ that appears most frequently across repetitions. In practice, $G$ is typically constrained to the range of $[3, 5]$ to balance hierarchical granularity with additional parameter overhead. We also report sensitivity analyses on $G$ in the experimental section, which demonstrate that DHAM remains robust with respect to the choice of hierarchy number.

## 4.2 HIERARCHICAL SOFT PROMPTS INJECTION

After obtaining the stage partition, we incorporate clustering-based semantic matching, in which trainable SPs provide stage-aware semantic guidance throughout the forward process.

Unlike conventional SP tuning that attaches prompts to every layer, our method injects prompts only at the representative layer of each cluster (see Fig. 5).

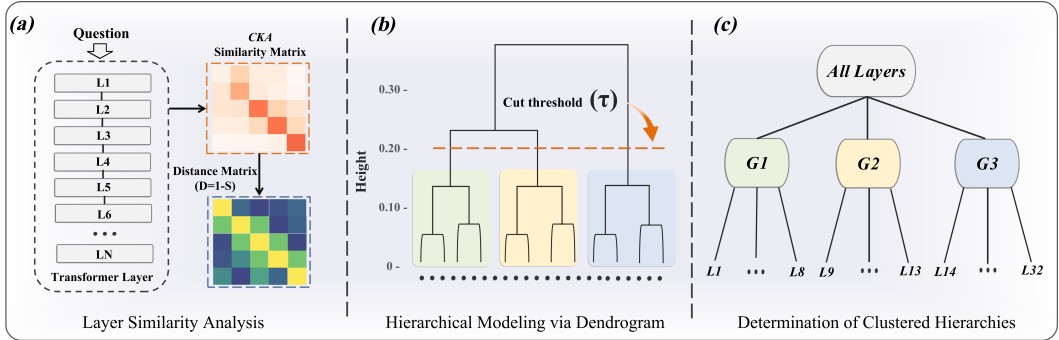

Figure 4: Overview of the hierarchical sharing procedure in DHAM. (a) Cross-layer similarity analysis: CKA is used to compute pairwise similarities between Transformer layers, forming a similarity matrix. (b) Hierarchical modeling via dendrogram: agglomerative clustering constructs a dendrogram that models the aggregation relations among layers. (c) Determination of clustered hierarchies: a cut threshold on the dendrogram yields candidate partitions, and the optimal number of hierarchies is selected via the Silhouette coefficient with bootstrap stabilization.

For each stage $\mathcal{G}_i$, we allocate a trainable soft prompt $P^{(i)} \in \mathbb{R}^{m \times d}$, where $m$ denotes the prompt length and $d$ the hidden dimension. The injection is implemented by concatenation along the sequence dimension, so that the prompt tokens are processed jointly with the original representation.

At the first layer, the input consists of raw embeddings $E_{\text{in}} \in \mathbb{R}^{n \times d}$. We concatenate it with the stage prompt $P^{(1)}$ to obtain:

$$X^{(1)} = \text{Concat}(E_{\text{in}}, P^{(1)}) \in \mathbb{R}^{(n+m) \times d} \qquad (5)$$

where the sequence length increases from $n$ to $n + m$ while the hidden dimension remains unchanged. For each subsequent stage $\mathcal{G}_i$, we similarly prepend the stage prompt $P^{(i)}$ to the input of its representative layer:

$$X^{(l)} = \text{Concat}(X^{(l)}, P^{(i)}) \in \mathbb{R}^{(n+m) \times d} \qquad (6)$$

This operation can be regarded as augmenting the sequence with $m$ "virtual tokens", thereby injecting stage-specific semantic information into the hidden space. Such a design enables the model to explicitly share prompts across stages, achieving stage-wise alignment and dynamic semantic control.

## 4.3 TRAINING OBJECTIVE

The optimization of DHAM follows the standard autoregressive language modeling task, with cross-entropy loss as the core objective. Given a target sequence $y_{1:n}$, the conditional probability at time step $t$ is defined as:

$$p_\theta(y_t \mid y_{<t}, X^{(L)}) \qquad (7)$$

where $X^{(L)}$ denotes the final representation after hierarchical soft prompt injection. The loss function is then defined as:

$$\mathcal{L}_{\text{CE}}(\theta) = -\sum_{t=1}^{n} \log p_\theta(y_t \mid y_{<t}, X^{(L)}) \qquad (8)$$

During training, we adopt the *teacher forcing* strategy, always conditioning on the ground-truth prefix $y_{<t}$ to ensure stable gradient propagation. Regarding parameter updates, we freeze the pretrained backbone parameters and only optimize the stage-specific soft prompts $\{P^{(i)}\}$ (as well as prompt length-related parameters, if learnable). This approach significantly reduces the number of trainable parameters, while ensuring that gradients flow effectively through $X^{(L)}$ to the prompts, thereby aligning the hierarchical structure with the downstream task objective. For clarity, we provide the detailed algorithmic workflow and pseudocode of DHAM in Appendix A.3.

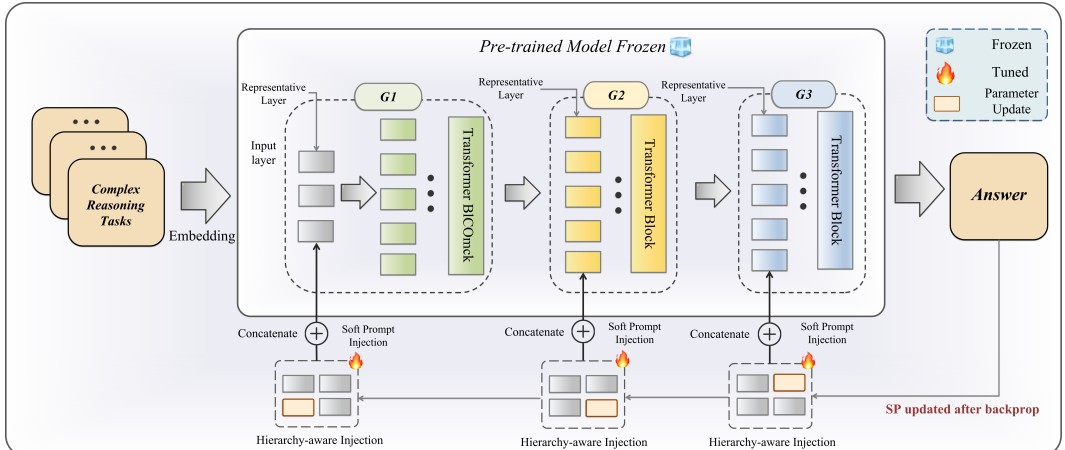

Figure 5: Dynamic stage-wise prompt injection in DHAM. At the representative layer of each stage, the input sequence is concatenated with a trainable soft prompt, which functions as virtual tokens to provide stage-specific semantic guidance. During training, the backbone is frozen, and only the soft prompt parameters (orange) are updated through backpropagation.

## 5 EXPERIMENT

In this section, we systematically conduct experiments to validate the effectiveness of our previous analysis and the proposed method. We design three types of evaluations: (1) performance comparisons across different models and reasoning benchmarks to assess the general applicability of our approach; (2) ablation studies on hierarchical partition strategies and the number of stages to analyze the impact of CKA-based clustering; and (3) visualization of information flow to intuitively demonstrate changes in inter-layer transmission patterns.

### 5.1 EXPERIMENTAL SETUP

**Models.** We evaluate the proposed DHAM method on four pretrained large language models of different scales and architectures: Llama-2-13B (Touvron et al., 2023), Llama-3-8B (Dubey et al., 2024), Mistral-7B (Chaplot, 2023), and DeepSeek-7B (Bi et al., 2024). These models cover diverse training corpora and reasoning capabilities, allowing us to comprehensively assess the generality and robustness of our approach.

**Datasets.** We consider three challenging reasoning benchmarks. GSM8K (Cobbe et al., 2021) is a large-scale grade-school math word problem dataset that primarily tests step-by-step reasoning and numerical calculation. MATH (Hendrycks et al., 2021) contains problems ranging from elementary algebra to advanced mathematics, focusing on multi-step logical reasoning and complex formula derivation. AQuA (Ling et al., 2017) is a multiple-choice dataset involving reasoning-chain integration and distractor discrimination. Together, these datasets cover distinct dimensions of reasoning, including step-by-step arithmetic, multi-step logic, and answer integration, thereby providing a comprehensive evaluation basis.

**Baselines.** We compare DHAM against several representative methods: (1) Pretrained model: directly using the frozen backbone without adaptation (2) Prompt tuning (Lester et al., 2021): injecting trainable prompt vectors at the input layer (3) Prefix tuning (Li & Liang, 2021): prepending trainable key–value vectors to each Transformer layer (4) LoRA (Hu et al., 2022): low-rank adaptation for efficient parameter tuning (5) DPC (Fan et al., 2025): dynamically detecting and masking harmful prompts during reasoning (6) DHAM (Ours): our proposed dynamic hierarchy-aware mechanism.

**Evaluation Metrics.** We evaluate model performance on GSM8K and AQuA using accuracy, defined as the proportion of predictions exactly matching the ground truth. For MATH, we adopt exact match (EM), which requires strict agreement with the reference solution at both the numeric and expression levels. All results are reported on the test set for fair comparison.

**Training Setup.** In all experiments, we freeze the backbone parameters and only train stage-specific soft prompts along with their associated weights. We adopt AdamW as the optimizer with a learning rate of $2 \times 10^{-5}$ and a batch size between 4 and 8, depending on GPU memory. We apply early stopping on the validation set to prevent overfitting.

## 5.2 PERFORMANCE EVALUATION

Table 1: Performance comparison across four models on GSM8K, MATH, and AQuA benchmarks. ("_" indicates that the result is not reported or not publicly available.)

| Method | Llama-2-13B | | | Llama-3-8B | | | Mistral-7B | | | DeepSeek-7B | | |
|---|---|---|---|---|---|---|---|---|---|---|---|---|
| | GSM8K | MATH | AQuA | GSM8K | MATH | AQuA | GSM8K | MATH | AQuA | GSM8K | MATH | AQuA |
| Pretrained model | 29.5 | 2.0 | 21.0 | 64.9 | 30.0 | 34.0 | 37.9 | 5.1 | 26.0 | 45.0 | 13.0 | 23.0 |
| Prompt tuning | 38.1 | 7.6 | 22.4 | 65.5 | 33.7 | 38.5 | 49.5 | 15.0 | 28.7 | 50.3 | 25.7 | 26.7 |
| Prefix tuning | 41.7 | 8.4 | 20.1 | 65.4 | 33.0 | 41.3 | 54.4 | 16.3 | 31.5 | 56.4 | 17.0 | 27.7 |
| ACT | 39.2 | 7.1 | 20.1 | 52.6 | 33.8 | 38.6 | 49.5 | 15.0 | 28.7 | – | – | – |
| LoRA | 12.7 | 7.4 | 24.8 | 40.9 | 27.1 | 42.9 | 45.1 | 12.4 | 26.4 | 45.0 | 26.3 | 27.0 |
| DPC | 41.9 | 9.2 | 31.1 | 67.6 | 36.3 | 42.5 | 51.1 | 16.4 | 31.9 | – | – | – |
| **DHAM(Ours)** | **43.8** | **9.7** | **33.4** | **74.0** | **38.9** | **44.7** | **57.5** | **18.1** | **34.7** | **60.1** | **28.7** | **30.9** |

Table 1 presents the performance of different tuning strategies across four representative LLMs on three reasoning benchmarks. Several observations can be made. First, compared with pretrained models, both prompt tuning and prefix tuning substantially improve accuracy, confirming the effectiveness of trainable prompt vectors in guiding reasoning. However, these static methods remain limited: their performance gains are mainly concentrated in the early stages of reasoning, while performance on more complex tasks, such as MATH and AQuA, remains insufficient. Second, LoRA yields unstable or even degraded results, with cases such as Llama-2-13B on GSM8K showing severe drops. This suggests that parameter-heavy adaptation methods are prone to overfitting and catastrophic forgetting in reasoning tasks. Third, DPC achieves some improvements by suppressing harmful prompts, but its effectiveness is inconsistent across models. In contrast, our proposed DHAM consistently outperforms all baselines across models and benchmarks. For example, DHAM improves Llama-2-13B on GSM8K from 29.5% to 43.8% and Llama-3-8B on MATH from 30.0% to 38.9%, while also delivering stable gains on Mistral-7B and DeepSeek-7B. These results demonstrate that DHAM exhibits strong generality and robustness. We hypothesize that the performance gains stem from DHAM's ability to mitigate stage misalignment and maintain stable cross-layer information transmission during reasoning, a hypothesis that we further validate in the following ablation studies.

## 5.3 ABLATION ON PARTITION STRATEGIES.

**Comparison of Hierarchical Partitioning Methods.**

Table 2 presents the experimental results of different hierarchical partitioning strategies and similarity metrics on Llama-3-8B across GSM8K, MATH, and AQuA benchmarks. Several observations emerge. First, using a single soft prompt without hierarchical partitioning yields the worst performance, indicating that treating all layers uniformly fails to exploit inter-layer differences. Introducing hierarchical partitioning consistently improves performance, and even the simplest uniform partition provides noticeable gains. Second, clustering-based strategies further enhance performance compared with uniform partitioning, suggesting that adaptively grouping layers according to representational similarity creates more meaningful stage boundaries. Among clustering metrics, cosine similarity and Euclidean distance offer moderate improvements, but their effectiveness remains limited. In contrast, CKA-based clustering achieves the best results, reaching 74.0% on GSM8K, 38.9% on MATH, and 44.7% on AQuA. This indicates that CKA more effectively captures cross-layer representational alignment, leading to more coherent stage formation and more stable information flow.

Table 2: Ablation study on different stage partition strategies and similarity metrics. Model: Llama-3-8B; benchmarks: GSM8K, MATH, and AQuA. Metric: Accuracy (Acc).

| Method | GSM8K | MATH | AQuA |
|---|---|---|---|
| Single SP (no partition) | 65.5 | 33.7 | 38.5 |
| Uniform partition | 65.8 | 33.9 | 39.0 |
| Cosine similarity + clustering | 66.9 | 34.2 | 40.5 |
| Euclidean distance + clustering | 66.5 | 34.0 | 40.1 |
| **CKA similarity + clustering (Ours)** | **74.0** | **38.9** | **44.7** |

Overall, the results confirm that the choice of partitioning strategy plays a positive role in mitigating stage misalignment. In particular, CKA-based clustering provides a more principled and effective hierarchical partitioning approach, maximizing the advantages of hierarchical soft prompting in complex reasoning tasks.

**Ablation on the Number of Hierarchical Stages.** To evaluate the impact of CKA-based hierarchical partitioning under different clustering thresholds, we conduct an ablation study on the number of stages $G$. Specifically, we adjust the clustering threshold to control the degree of layer merging, which results in different hierarchical structures. Varying $G$ only changes the distribution of prompts across the hierarchy, while keeping the overall prompt budget fixed.

The results in Table 3 show that a moderate number of stages yields the best performance, while too few or too many reduce accuracy. Llama-2-13B performs best at $G = 4$ (43.8%), Llama-3-8B at $G = 3$ (74.0%) but drops sharply with larger $G$, and Mistral-7B and DeepSeek-7B at $G = 5$ (57.5%) and $G = 3$ (60.1%), respectively, reflecting model-specific optima.

Table 3: Ablation study on the number of hierarchical stages $G$ across different models. The total prompt token budget is fixed at 64 to ensure a fair comparison. Metric: Accuracy (Acc).

| Model | Stages $G$ | | | | | | |
|---|---|---|---|---|---|---|---|
| | $G = 2$ | $G = 3$ | $G = 4$ | $G = 5$ | $G = 6$ | $G = 7$ | $G = 8$ |
| Llama-2-13B | 41.0 | 43.0 | **43.8** | 42.9 | 42.0 | 41.6 | 40.8 |
| Llama-3-8B | 73.0 | **74.0** | 72.0 | 65.0 | 69.0 | 69.0 | 67.0 |
| Mistral-7B | 54.8 | 55.9 | 56.8 | **57.5** | 57.0 | 56.2 | 55.1 |
| DeepSeek-7B | 58.2 | **60.1** | 59.1 | 58.0 | 57.2 | 56.5 | 55.9 |

In summary, too few stages merge layers and obscure hierarchical distinctions, while too many stages fragment prompts and disrupt information flow. These results support our hypothesis that aligning hierarchy with model depth is essential: an appropriate stage number stabilizes information flow, prevents misalignment from over-aggregation or over-segmentation, and maximizes reasoning performance under a fixed prompt budget.

## 5.4 VISUALIZATION

We analyze the information flow under DHAM. Figure 6 shows that layer-wise saliency follows a unimodal and smooth rise-and-fall pattern, which aligns with the stable information flow hypothesis: saliency gradually shifts from prompts to the problem and intermediate reasoning steps, thereby forming continuous and directional transmission across layers. In contrast, multiple peaks or sharp reversals in the middle and later stages indicate stage misalignment, where prompts compete with task signals and disrupt transmission. Thus, a smooth unimodal curve corresponds to stable reasoning, whereas fluctuating multi-peak patterns reflect misalignment.

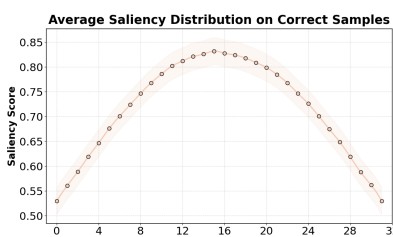

Figure 6: Average layer-wise saliency distribution on correctly solved GSM8K cases under our method DHAM (G=3).

## 6 CONCLUSION

In this paper, we show through saliency analysis that effective information flow in large-model reasoning follows a smooth, unimodal migration, where information passes layer by layer and gradually concentrates on the problem and intermediate steps. In practice, however, models often exhibit multi-peaked oscillations that cause backflow and weaken reasoning. To address this, we propose the Dynamic Hierarchy-Aware Mechanism (DHAM), which uses CKA-based hierarchical partitioning and stage-specific prompt regulation to guide reasoning at appropriate depths. Experiments demonstrate that DHAM restores smooth cross-layer flow, mitigates disruption, and significantly improves accuracy on complex reasoning tasks.

## 7 REPRODUCIBILITY STATEMENT

We place strong emphasis on the reproducibility of our work and provide multi-level support across the main paper, appendix, and supplementary materials. In particular, Section 4 describes the proposed Dynamic Hierarchy-Aware Mechanism (DHAM) in detail, including CKA-based inter-layer similarity computation, hierarchical clustering criteria, and stage-wise soft prompt injection. To further reduce implementation barriers, Appendix A.3 provides full pseudocode for the core algorithms (stage partitioning, prompt injection, and training), and explains the complete workflow step by step to ensure transparency and operability. Section 5 systematically introduces the experimental setup, including models (Llama-2-13B, Llama-3-8B, Mistral-7B, DeepSeek-7B), datasets (GSM8K, MATH, AQuA), training configurations (optimizer, learning rate, batch size, freezing strategy), and evaluation metrics, ensuring that experimental conditions are clearly documented. Additional analyses are reported in Section 5 and Appendix A.2, covering ablations on partition strategies and the number of stages $G$, as well as visualizations of information flow across layers. These supplementary results support the robustness of our method and provide practical guidance for replication. All datasets used in this work are publicly available, and their sources are clearly cited in the main text. Upon acceptance, we will release the full source code and training scripts to further facilitate replication and extension of our research by the community.

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

# A APPENDIX

## A.1 THE USE OF LARGE LANGUAGE MODELS (LLMS)

The authors utilized OpenAI's GPT-5 to improve the grammar, clarity, and conciseness of the text. All scientific contributions, methodology design, experiments, and analyses are the original work of the authors, who take full responsibility for the paper's content.

## A.2 ANALYSIS OF STAGE MISALIGNMENT IN SOFT PROMPTS

As shown in Figure 7, different insertion stages lead to distinct patterns of saliency propagation. Without SP (a), the trajectory exhibits oscillatory fluctuations, suggesting unstable information flow. Early insertion at Layer 1 (b) produces a unimodal rise–fall curve, indicating a smooth migration of information across depth and a more coherent flow. By contrast, mid- (c) and late-stage (d) insertions yield multi-peaked and highly oscillatory trajectories, with frequent spikes and reversals that disrupt stability. These results align with our hypothesis that early prompts facilitate stable inter-layer propagation, whereas later prompts induce stage misalignment and interfere with reasoning.

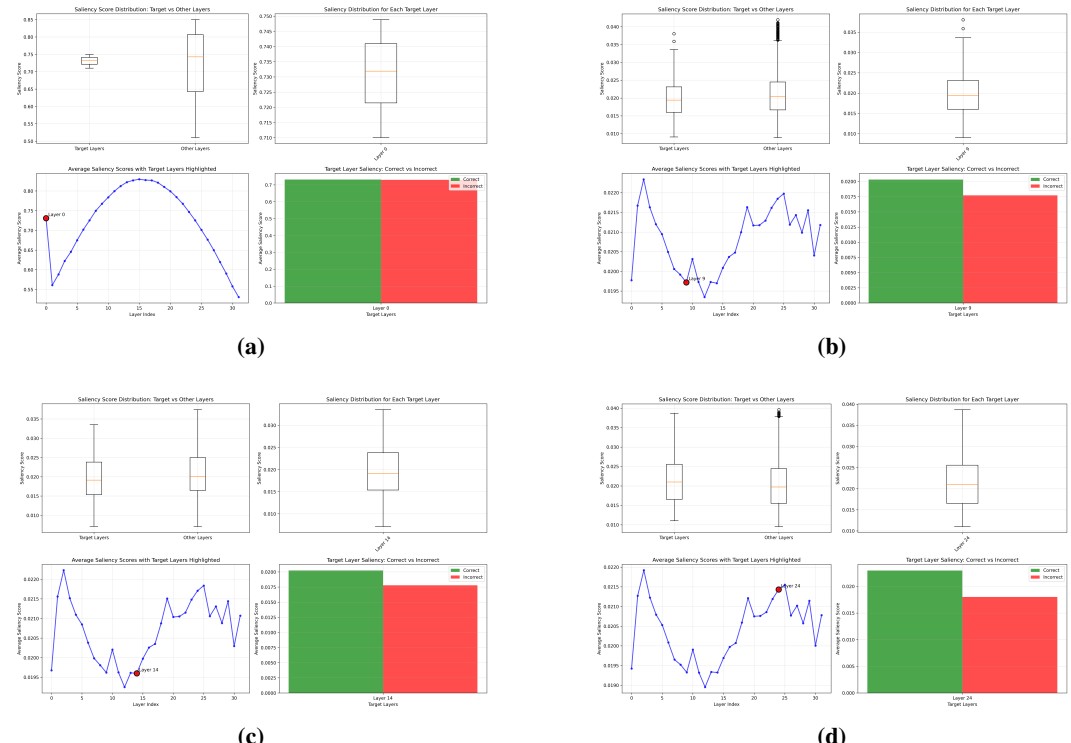

Figure 7: Layer-wise saliency analysis under different soft prompt (SP) insertion stages. (a) w/o baseline, (b) early insertion at Layer 1, (c) mid-stage insertion at Layer 9, and (d) late-stage insertion at Layer 24. Each panel reports saliency distributions, average trajectories, and correctness comparison, highlighting how insertion stage systematically shapes information flow.

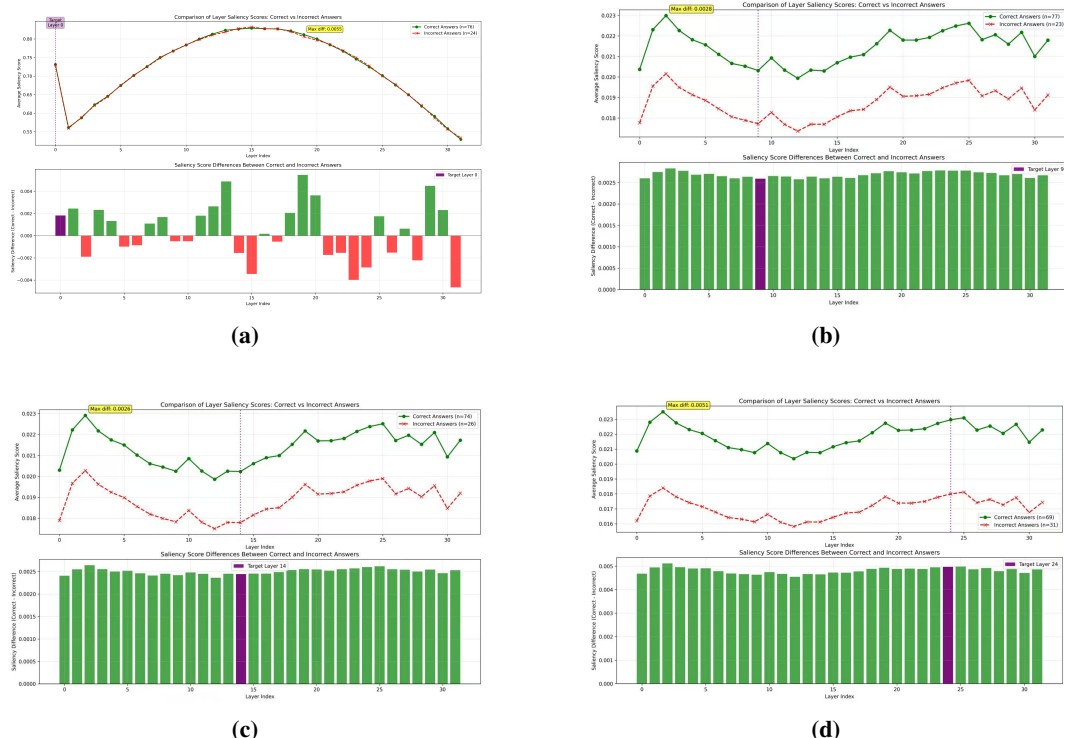

Figure 8: Overall comparison of saliency trajectories between correct and incorrect predictions under different SP insertion stages: (a) Layer 1, (b) Layer 9, (c) Layer 14, and (d) Layer 24. Each panel shows average saliency scores and their differences across layers.

As illustrated in Figure 8, the gap between correct and incorrect samples varies systematically with the insertion stage. When SPs are inserted at Layer 1 (a), correct and incorrect trajectories largely overlap, with only small differences concentrated in shallow layers, suggesting that early insertion encourages a stable and consistent propagation path. By contrast, mid- (b, c) and late-stage (d) insertions yield more pronounced and persistent gaps between correct and incorrect samples, indicating that saliency is increasingly diverted from task-relevant signals. This comparison further confirms that early-stage prompts promote more robust alignment between saliency propagation and reasoning correctness, whereas later-stage prompts exacerbate stage misalignment.

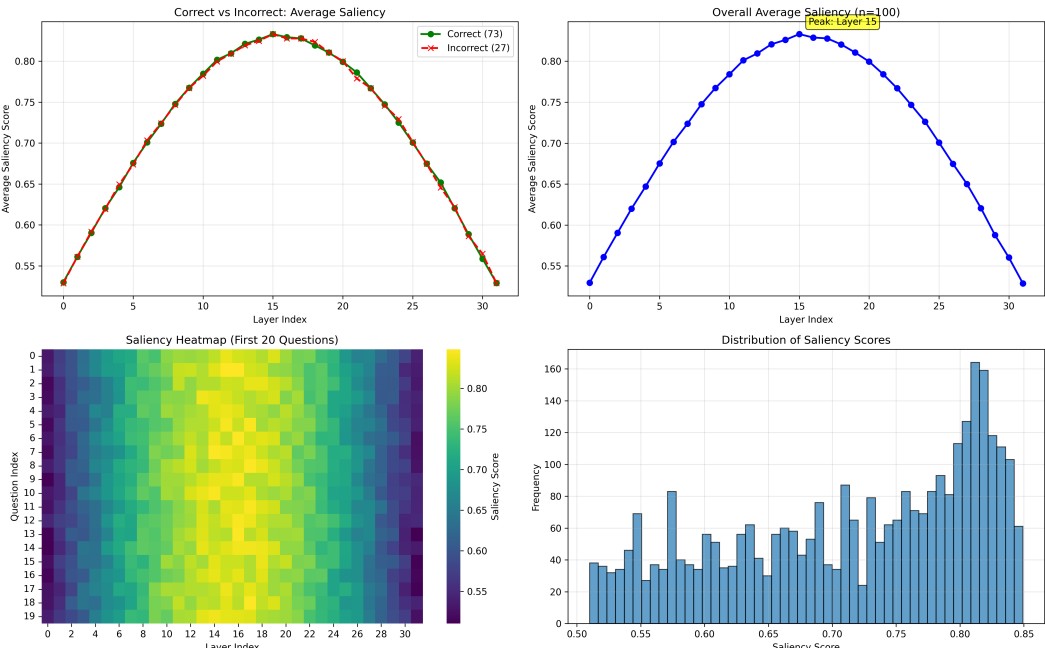

Figure 9: Comprehensive saliency statistics across layers. (Top-left) Average saliency trajectories for correct vs. incorrect predictions. (Top-right) Overall average saliency with peak at Layer 15. (Bottom-left) Heatmap of saliency evolution for 20 representative samples. (Bottom-right) Histogram of saliency score distribution.

Figure 9 provides an overall view of saliency behavior. The average trajectories for correct and incorrect samples (top-left) nearly overlap, suggesting that global saliency trends are largely consistent regardless of correctness. The overall average curve (top-right) exhibits a clear unimodal pattern with a peak around Layer 15, indicating that information concentration emerges in mid-depth layers. The heatmap of representative samples (bottom-left) further confirms this trend, showing gradual migration of saliency from shallow to mid layers. Finally, the histogram (bottom-right) reveals that most saliency scores are concentrated in the 0.70–0.82 range, demonstrating stable activation magnitudes across layers. Together, these results suggest that saliency exhibits a universal depth-dependent trajectory, with mid-layer concentration acting as a key stage in information propagation.

## A.3 PSEUDOCODE

This section provides the detailed pseudocode for the primary components of the DHAM framework.

---

**Algorithm 1 CKA-based Hierarchical Stage Partitioning**

---

**Require:** Pretrained LLM $\mathcal{M}$ with $L$ Transformer layers; unlabeled calibration set $\mathcal{C}$; cut-threshold candidates $\{\tau\}$; bootstrap rounds $B$

**Ensure:** Stage groups $\{\mathcal{G}_i\}_{i=1}^{G}$; representative-layer map $r(i)$ for each stage $i$

    *# Collect layer-wise representations on calibration set*

1: **for** each sequence $x \in \mathcal{C}$ **do**
2:     Run forward pass of $\mathcal{M}$ on $x$ and cache hidden states $\{X^{(l)}(x) \in \mathbb{R}^{n_x \times d}\}_{l=1}^{L}$
3: **end for**
    *# Build CKA similarity matrix*
4: Initialize $S \in [0,1]^{L \times L}$
5: **for** $l = 1$ **to** $L$ **do**
6:     **for** $l' = 1$ **to** $L$ **do**
7:         Form $X = \text{stack}_{x \in \mathcal{C}}(X^{(l)}(x)), Y = \text{stack}_{x \in \mathcal{C}}(X^{(l')}(x))$
8:         $K \leftarrow XX^{\top}, Lmat \leftarrow YY^{\top}, H \leftarrow I - \frac{1}{n}\mathbf{1}\mathbf{1}^{\top}$
9:         $K \leftarrow HKH, L \leftarrow HLmatH$                $\triangleright$ centered kernels
10:         $\text{HSIC}(K,L) \leftarrow \text{tr}(K^{L)})$
11:         $S[l,l'] \leftarrow \frac{\text{HSIC}(K,L)}{\sqrt{\text{HSIC}(K,K) \cdot \text{HSIC}(L,L)}}$
12:     **end for**
13: **end for**
    *# Agglomerative clustering and model selection*
14: Build dendrogram $\mathcal{T}$ from $S$
15: **for** each $\tau$ in $\{\tau\}$ **do**
16:     Obtain partition $\{\mathcal{G}_i(\tau)\}$ by cutting $\mathcal{T}$ at $\tau$
17:     Compute Silhouette score $\text{Sil}(\tau)$
18: **end for**
19: $\tau^{\star} \leftarrow \arg\max_{\tau} \text{Sil}(\tau)$
20: **(Bootstrap)** Repeat lines 12–17 for $B$ resamples of $\mathcal{C}$ and pick the most frequent $G$
21: $\{\mathcal{G}_i\}_{i=1}^{G} \leftarrow$ partition at $\tau^{\star}$ (or bootstrap majority)
    *# Choose representative layer for each stage*
22: **for** $i = 1$ **to** $G$ **do**
23:     $r(i) \leftarrow \min \mathcal{G}_i$                $\triangleright$ default: shallowest layer in stage $i$
24: **end for**
25: **return** $\{\mathcal{G}_i\}_{i=1}^{G}, r(i)$

---

**Algorithm 2 DHAM Forward: Stage-wise Soft Prompt Injection**

---

**Require:** Token embeddings $E_{\text{in}} \in \mathbb{R}^{n \times d}$; stage groups $\{\mathcal{G}_i\}_{i=1}^{G}$; representative-layer map $r(i)$; stage prompts $\{P^{(i)} \in \mathbb{R}^{m \times d}\}$; pretrained backbone $\mathcal{M}$ (frozen)

**Ensure:** Final hidden state $X^{(L)}$

1: $Z^{(1)} \leftarrow E_{\text{in}}$
2: **for** $\ell = 1$ **to** $L$ **do**
3:     **if** $\ell = r(i)$ for some stage $i$ **then**
4:         $Z^{(\ell)} \leftarrow \text{Concat}_{\text{seq}}(Z^{(\ell)}, P^{(i)})$     $\triangleright$ concatenation along sequence dim $(n \rightarrow n + m)$
5:     **end if**
6:     $X^{(\ell)} \leftarrow \text{TransformerLayer}_{\ell}(Z^{(\ell)})$         $\triangleright$ frozen weights
7:     **if** $\ell < L$ **then**
8:         $Z^{(\ell+1)} \leftarrow X^{(\ell)}$
9:     **end if**
10: **end for**
11: **return** $X^{(L)}$

---

---

**Algorithm 3 Training DHAM with Cross-Entropy (Teacher Forcing)**

---

**Require:** Training set $\mathcal{D} = \{(x, y)\}$; stage groups $\{\mathcal{G}_i\}$; representative layers $r(i)$; prompts $\{P^{(i)}\}$ (trainable); pretrained backbone $\mathcal{M}$ and LM head (frozen); optimizer $\mathcal{O}$; learning rate $\eta$

**Ensure:** Trained stage prompts $\{P^{(i)}\}$

1: Freeze all parameters of $\mathcal{M}$ and LM head; set `requires_grad=False` except $\{P^{(i)}\}$
2: **for** each minibatch $\mathcal{B} \subset \mathcal{D}$ **do**
3:     $\mathcal{L} \leftarrow 0$
4:     **for** each $(x, y)$ in $\mathcal{B}$ **do**
5:         $E_{\text{in}} \leftarrow \text{Embed}(x)$
6:         $X^{(L)} \leftarrow \text{DHAM-FORWARD}(E_{\text{in}}, \{\mathcal{G}_i\}, r(\cdot), \{P^{(i)}\}, \mathcal{M})$
7:         $\mathcal{L} \leftarrow \mathcal{L} - \sum_{t=1}^{|y|} \log p_\theta(y_t \mid y_{<t}, X^{(L)})$
8:     **end for**
9:     Compute gradients $\nabla_{\{P^{(i)}\}} \mathcal{L}$
10:     Update prompts: $\{P^{(i)}\} \leftarrow \mathcal{O}(\{P^{(i)}\}, \nabla, \eta)$
11:     Zero optimizer/memory buffers
12: **end for**
13: **return** $\{P^{(i)}\}$

---

### A.3.1 PSEUDOCODE EXPLANATION

To complement the pseudocode presented in Algorithms 1–3, we provide a step-by-step explanation of the DHAM workflow. The entire pipeline consists of three key components: stage partitioning, stage-wise soft prompt injection, and optimization.

**Stage Partitioning (Algorithm 1).** The first step is to analyze representational similarity across layers of the pretrained LLM. We collect hidden states from a small calibration set and compute pairwise Centered Kernel Alignment (CKA) scores to quantify distributional similarity between layers. These scores form a similarity matrix, which is then passed to agglomerative hierarchical clustering to generate a dendrogram that reflects cross-layer functional relationships. By sweeping over multiple cut thresholds $\tau$ and evaluating the resulting partitions with the Silhouette score, we determine the optimal number of hierarchies $G$. To ensure stability, bootstrap resampling is applied and the most frequent partition is selected. Finally, each stage is assigned a representative layer, typically the shallowest layer in the cluster, which will later serve as the insertion point for stage prompts.

**Stage-wise Prompt Injection (Algorithm 2).** Once stages are determined, we introduce trainable soft prompts at their representative layers. Unlike conventional prompt tuning that attaches prompts to every layer, our method concatenates a stage-specific prompt only at the designated entry point of each stage. Concretely, for stage $\mathcal{G}_i$, its prompt $P^{(i)}$ is prepended to the sequence at $r(i)$, expanding the sequence length from $n$ to $n + m$ while keeping hidden dimension $d$ unchanged. The modified input is then passed through the frozen Transformer backbone, allowing the injected virtual tokens to steer the information flow in a stage-aware manner. This design enables dynamic guidance across stages while avoiding over-saturation of prompts in deeper layers.

**Training Objective (Algorithm 3).** During training, we freeze all parameters of the pretrained backbone and LM head, optimizing only the stage-specific prompts $\{P^{(i)}\}$. We adopt teacher forcing, where at each time step the model conditions on the ground-truth prefix $y_{<t}$ to predict the next token $y_t$. The objective is the standard cross-entropy loss over the sequence. Gradients are propagated through the frozen backbone to the prompt embeddings, which are updated using AdamW or a similar optimizer. This reduces the number of trainable parameters by several orders of magnitude, while still allowing the injected prompts to adapt to task-specific reasoning requirements.

In summary, the three algorithms together describe the full DHAM pipeline: first partitioning the model into semantically coherent stages, then injecting stage-aware soft prompts at the representative layers, and finally training these prompts with cross-entropy under teacher forcing. This modular pseudocode reflects the structural motivation of DHAM, making its implementation transparent and reproducible.

## A.4 VISUALIZATION

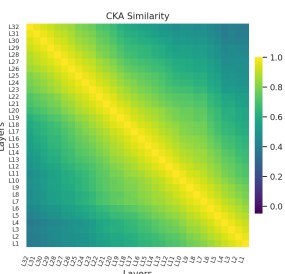 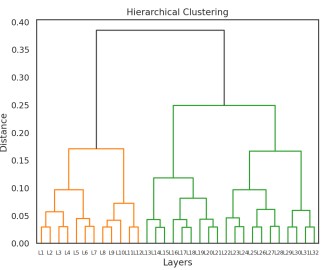 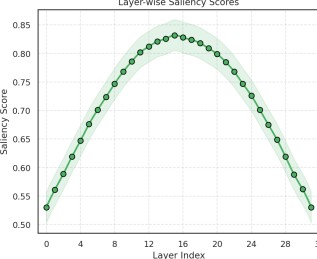

Figure 10: CKA-based hierarchical clustering pipeline. (Left) Layer-wise representational similarity measured by CKA. (Middle) Agglomerative hierarchical clustering produces a dendrogram that models aggregation relations across layers. (Right) Layer-wise saliency scores are aligned with the derived hierarchy to guide stage partitioning.

As illustrated in Figure 10, the hierarchical partitioning process follows a three-step pipeline. First, CKA is employed to compute pairwise similarity across Transformer layers, yielding a structured similarity matrix. Second, agglomerative hierarchical clustering constructs a dendrogram that captures aggregation relations and potential stage boundaries. Finally, the obtained partitions are aligned with saliency-based layer importance, providing a principled basis for stage-wise prompt injection. This integration ensures that the partitions are both representation-driven and task-aware, stabilizing cross-layer organization.

## A.5 NOTATION

Table 4: Table of Mathematical Symbols

| Symbol | Meaning |
|---|---|
| $L$ | Number of Transformer layers in the backbone model |
| $d$ | Hidden dimension of Transformer representations |
| $n$ | Length of the input sequence (number of tokens) |
| $m$ | Length of each trainable soft prompt (number of virtual tokens) |
| $X^{(l)} \in \mathbb{R}^{n \times d}$ | Hidden representation at the $l$-th Transformer layer |
| $E_{\text{in}} \in \mathbb{R}^{n \times d}$ | Input token embeddings |
| $P^{(i)} \in \mathbb{R}^{m \times d}$ | Trainable soft prompt for stage $\mathcal{G}_i$ |
| $\{\mathcal{G}_i\}_{i=1}^G$ | Partition of Transformer layers into $G$ hierarchical stages |
| $r(i)$ | Representative layer index of stage $\mathcal{G}_i$ |
| $S \in [0,1]^{L \times L}$ | CKA similarity matrix across all layers |
| $\tau$ | Cut threshold applied to the dendrogram to obtain $G(\tau)$ clusters |
| $G(\tau)$ | Number of hierarchies obtained at threshold $\tau$ |
| $G^\star$ | Optimal number of hierarchies chosen via Silhouette score |
| $\mathcal{C}$ | Calibration set used to compute CKA similarity |
| $\mathcal{D} = \{(x,y)\}$ | Training dataset (input $x$ with target sequence $y$) |
| $X^{(L)}$ | Final hidden state of the model after $L$ layers (input to LM head) |
| $p_\theta(y_t \mid y_{<t}, X^{(L)})$ | Conditional probability of predicting token $y_t$ |
| $\mathcal{L}_{\text{CE}}$ | Cross-entropy loss for autoregressive language modeling |
| $B$ | Number of bootstrap rounds for stabilizing clustering |
| $\eta$ | Learning rate for optimizing prompts |

## A.6  FIGURE

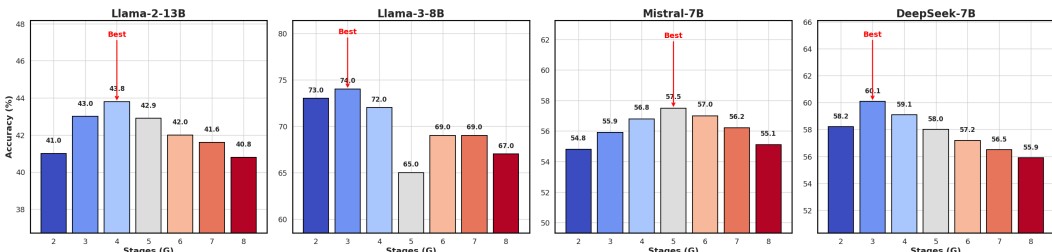

Figure 11: Ablation on the number of hierarchical stages $G$ across different LLMs. Each subfigure corresponds to one backbone model: Llama-2-13B, Llama-3-8B, Mistral-7B, and DeepSeek-7B. The $x$-axis denotes the number of hierarchical stages $G$, while the $y$-axis reports task accuracy. Bars are color-coded by stage number, and the best-performing configuration is highlighted with a red arrow. Results demonstrate that performance exhibits a unimodal trend: too few stages fail to capture sufficient hierarchy, while too many stages dilute semantic guidance and reduce stability.

**Ablation on Hierarchical Stages.** Figure 11 presents the effect of varying the number of hierarchical stages $G$ under a fixed prompt budget. We observe that performance is sensitive to the choice of $G$, typically following a unimodal distribution. When $G$ is too small (e.g., $G = 2$), the clustering collapses multiple functionally distinct layers into a single stage, which limits the ability of stage-specific prompts to provide fine-grained guidance. Conversely, when $G$ is too large (e.g., $G \geq 6$), the prompts become fragmented across stages, weakening semantic consistency and increasing optimization difficulty. Moderate values of $G$ (between 3 and 5) consistently yield the best results across all evaluated models, suggesting that DHAM benefits from a balanced hierarchical granularity that matches the intrinsic layer organization of LLMs.

## A.7  FUTURE WORK.

While our ablation results suggest that moderate stage numbers ($3 \leq G \leq 5$) strike a good balance between semantic granularity and parameter efficiency, further exploration is warranted. One promising direction is to make the stage partitioning process *adaptive*, allowing $G$ to vary dynamically across tasks, datasets, or even input instances. Another avenue is to jointly optimize the partitioning and prompt parameters in an end-to-end manner, rather than precomputing the hierarchy. Such adaptive and task-aware extensions could further enhance the generality and robustness of DHAM.

