# OpenReview forum: "Unlocking Coherent Reasoning in LLMs with Hierarchical Soft Prompts"
_ICLR.cc/2026/Conference — ICLR 2026 Conference Withdrawn Submission_

### Official Review · Reviewer_4SFv · 2025-10-26

**Soundness:** 2
**Presentation:** 2
**Contribution:** 2
**Rating:** 4
**Confidence:** 4

**Summary:**

This work found that previous soft prompts often disrupted information flow and reduced reasoning. They argue that soft prompts should not be limited to the activation and guidance stages but should be inserted into appropriate stages to ensure smooth information flow between layers. Therefore, they proposed a Dynamic Hierarchical Awareness Mechanism (DHAM) to ensure effective coordination between the various stages of reasoning.

**Strengths:**

The authors innovatively employ saliency scores to analyze and visualize the internal information flow within the LLM during reasoning. The "Stable Information Flow Hypothesis"  derived from this analysis provides an insightful foundation for the paper's methodology. This approach of using information flow diagnostics to guide prompt design offers a fresh perspective on understanding and optimizing the model's reasoning process

**Weaknesses:**

1. The DHAM method can essentially be viewed as a sparse special case of Prefix Tuning, simplifying from injecting prompts at *every* layer to injecting at only a *subset* of layers. Consequently, this feels less like a new mechanism and more like a structural pruning of an existing framework.

2. The paper reports that DHAM consistently and significantly outperforms Prefix Tuning. This result is counter-intuitive, given that Prefix Tuning theoretically offers stronger layer-by-layer control. The authors do not explicitly state whether all baseline methods were fairly reproduced under the exact same experimental configuration.

3. There appears to be a disconnect between the preliminary analysis and the final method. The analysis in Section 3 concludes that early injection (Layer 1) produces the most ideal information flow. However, the proposed DHAM method adopts a multi-stage injection strategy. The authors need to provide a clearer explanation for why they abandoned the optimal single-point solution (identified in their own analysis) in favor of a more complex multi-point scheme.

4. The paper's interpretation of the "saliency score" is overly vague. The process of compressing complex layer information into a "single scalar" lacks transparency. More importantly, based on prior work, saliency scores reflect the *strength* of information interaction, not its *quality*. The authors' assumption that "smooth" curves equate to "better" reasoning and "oscillations" represent "interference" is an unsubstantiated hypothesis.

6.The caption for Figure 2 describes it as calculating the "change between consecutive layers (later minus earlier)", which should produce a one-dimensional (1D) result. However, the figure itself is presented as a two-dimensional (L x L) heatmap. This contradiction between the description and the format makes it impossible to understand what the "smooth diagonal band" or "scattered hotspots" actually represent.

**Questions:**

see weakness

---

### Official Review · Reviewer_Sgtt · 2025-10-30

**Soundness:** 2
**Presentation:** 2
**Contribution:** 3
**Rating:** 4
**Confidence:** 4

**Summary:**

This paper investigates the role of soft prompt tuning in improving reasoning performance of large language models (LLMs). While previous works show that soft prompts can effectively activate prior knowledge and facilitate early reasoning, this paper observes that maintaining strong prompt influence in later reasoning stages can disrupt information flow and degrade performance. To address this issue, the paper proposes a Dynamic Hierarchy-Aware Mechanism (DHAM) that dynamically regulates soft prompts across reasoning stages. Specifically, DHAM performs hierarchical clustering to identify stage-specific representations and adaptively activates soft prompts based on semantic alignment, thereby ensuring smoother and more coherent information transmission through model layers. Experimental results demonstrate consistent improvements across different models and reasoning benchmarks. Ablation studies suggest that using CKA-based clustering and a moderate number of reasoning stages achieves the best performance, supporting the paper’s hypothesis of stable information flow as a key factor for effective reasoning.

**Strengths:**

1. This paper provides both a visualization and a quantitative analysis of the inter-layer information flow in existing methods, which effectively validates the rationality of the proposed motivation.
2. The experiments cover multiple large language models, including Llama-2-13B, Llama-3-8B, Mistral-7B, and DeepSeek-7B, and evaluate the proposed method across various benchmarks, demonstrating its effectiveness and generalizability.
3. The visualizations used in this paper to represent the flow and transmission of information are very clear, effectively compensating for areas where the author's textual descriptions are less precise.

**Weaknesses:**

1. This paper is quite difficult to understand, especially the Introduction section. In this part, phrases like in other words are used repeatedly. Upon closer examination, the explanations following these phrases are relatively clear, but the preceding parts remain very abstract. This paper would benefit from careful revision and polishing by a native English speaker. In addition, the paper repeatedly uses the terms stage and layer, but from the descriptions, they seem to refer to the same concept, which can be easily confusing.
2. This paper emphasizes the use of a hierarchical soft prompt, but the hierarchy is only reflected in dividing the model’s layers using a clustering method, and then applying soft prompts to each group. It is questionable whether this approach can truly be considered hierarchical, as the design does not establish meaningful hierarchical relationships beyond the arbitrary clustering of layers.
3. The ablation study in this paper shows that the Number of Hierarchical Stages performs best when set to 3 or 4. However, it is not clear which layers are included in each stage for different large models, nor whether the number of layers per stage is consistent across different models.

**Questions:**

1. Looking at Figure 4(c), the layers included in G1, G2, and G3 are all consecutive. Is it possible for the stages to contain interleaved layers instead?

---

### Official Review · Reviewer_ELDL · 2025-10-31

**Soundness:** 2
**Presentation:** 2
**Contribution:** 3
**Rating:** 4
**Confidence:** 4

**Summary:**

This paper identifies that static soft prompts (SP) can disrupt information flow when injected into middle or late layers. To address this, the paper proposes the Dynamic Hierarchy-Aware Mechanism (DHAM), which uses CKA-based clustering to group layers into functional stages and injects distinct prompts at each stage. This hierarchical alignment is shown to stabilize information flow and improve reasoning performance.  However, clearer experimental evidence should be provided.

**Strengths:**

- Originality: The paper's core insight is novel: that effective prompt tuning must align with the hierarchical reasoning process, not just inject static prompts. This shift from static injection to hierarchy-aware alignment is a significant contribution.
- Clarity: Figures 4 and 5 provide a clear and intuitive illustration of the DHAM algorithm's core concepts and workflow.
- Quality: The empirical observations motivating the work are interesting.
- Significance: The method and analysis proposed in the paper are simple and effective, and easy to reuse.

**Weaknesses:**

1. The paper's central claim is that mid- or late-layer SP injection induces "repeated backflows and spikes." However, the oscillatory patterns in Fig 1(c) (Layer 9) and 1(d) (Layer 24) appear qualitatively similar. This makes it difficult to assess whether the desired "unimodal and smooth migration of saliency" is a continuous property (i.e., more smooth = better accuracy) or a sudden threshold effect. The link between the degree of smoothness and the degree of performance improvement is not sufficiently established.
2. The paper lacks crucial implementation details for the baseline methods. Key hyperparameters, such as the LoRA rank and alpha, or the prompt length for Prompt Tuning, are not specified. This information is essential for evaluating the fairness of the comparison and for ensuring reproducibility.
3. The set of baselines, while including classics, is somewhat limited and dated. Most baselines (except DPC) are from 2022 or earlier. A comparison against more recent and advanced PEFT methods would provide a stronger validation. Optional comparisons include:
   - Wu, Zhengxuan, et al. "Reft: Representation finetuning for language models." Advances in Neural Information Processing Systems 37 (2024): 63908-63962.
   - Wang, Shaowen, Linxi Yu, and Jian Li. "Lora-ga: Low-rank adaptation with gradient approximation." Advances in Neural Information Processing Systems 37 (2024): 54905-54931.
   - Shi, Z., and A. Lipani. "DePT: Decomposed Prompt Tuning for Parameter-Efficient Fine-tuning." 12th International Conference on Learning Representations, ICLR 2024. Vol. 2024. International Conference on Learning Representations (ICLR), 2024.
   - Lan, Pengxiang, et al. "Efficient and Effective Prompt Tuning via Prompt Decomposition and Compressed Outer Product." Proceedings of the 2025 Conference of the Nations of the Americas Chapter of the Association for Computational Linguistics: Human Language Technologies (Volume 1: Long Papers). 2025.
4. The method's generality could be further strengthened by validating it on other powerful, newer models, such as the Qwen2 and Qwen3 series.

**Questions:**

1. In Figure 1, SP injection at Layer 9 and Layer 24 produces oscillatory patterns similar to the pretrained model (w/o SP). However, in Figure 3, these two trained configurations (Accuracy: 62.7% and 58.3%) perform worse than the pretrained model (Accuracy: 67.3%). This seems counter-intuitive. If the information flow patterns are similarly "disrupted," why does adding a trainable prompt result in such a severe performance degradation compared to having no prompt at all?
2. My most significant question: following Weakness 1, could the authors provide a more quantitative analysis linking information flow stability to accuracy? For example, could a metric for "smoothness" (e.g., the variance of the saliency curve, or number of local peaks) be plotted against the final task accuracy to provide more direct evidence that a "unimodal and smooth" flow leads to better performance?
3. Other questions are covered in the Weaknesses section.

---

### Official Review · Reviewer_h7to · 2025-11-03

**Soundness:** 2
**Presentation:** 3
**Contribution:** 2
**Rating:** 4
**Confidence:** 3

**Summary:**

This paper proposes a novel method called Dynamic Hierarchical Awareness Mechanism (DHAM), which aims to address the issues of incoherent information flow and performance degradation in large language models (LLMs) during complex reasoning tasks due to the static injection of soft prompts. The authors found through analysis that improper prompt injection can cause severe oscillations in information propagation between model layers, disrupting the coherence of reasoning. To this end, DHAM first automatically divides the model's Transformer layers into several functionally similar semantic stages using Centered Kernel Alignment (CKA) and hierarchical clustering. Subsequently, it injects trainable soft prompts only at the starting layers of each stage, achieving phased and dynamic guidance of the information flow. Experiments show that this stage-aware injection strategy, especially the injection in the early stages, can effectively promote the smooth transfer of information and significantly improve the model's accuracy on complex reasoning tasks such as GSM8K and MATH.

**Strengths:**

1. Originality: The paper is the first to systematically diagnose the issue of incoherent information flow in LLM reasoning. By visualizing the propagation trajectory of inter-layer significance, it reveals the phenomenon of severe fluctuation in significance scores across different layers. The proposed DHAM method innovatively combines hierarchical clustering with soft prompt injection, offering a new approach to enhancing the reasoning capabilities of LLMs.
2. Experimental Quality: The experimental design is comprehensive and rigorous. It evaluates four models of different scales and architectures (Llama-2-13B, Llama-3-8B, Mistral-7B, DeepSeek-7B) using three key reasoning benchmarks (GSM8K, MATH, AQuA).
3. Clarity: The paper is logically structured and provides detailed descriptions of the methods (including core algorithms such as CKA-based inter-layer similarity computation and hierarchical clustering criteria). The figures and tables are professional and information-rich, effectively supporting the main arguments.
4. Significance: It addresses a key practical issue in LLMs during complex reasoning tasks. The proposed DHAM method significantly improves model performance on datasets such as GSM8K and MATH, holding certain value for both LLM reasoning research and practical applications.

**Weaknesses:**

1. Insufficient theoretical depth: The paper is mainly based on experimental observations and verifications, lacking in-depth theoretical explanations of the relationship between incoherent information flow and decreased reasoning performance. In particular, as to why injection in the early stages works best, the paper only explains through experimental phenomena without providing a theoretical basis.
2. Insufficient experimental details: The specific parameters for CKA computation (such as kernel function selection and hyperparameter settings) are not clearly stated. Although four models were used, only three sets of benchmarks were selected. It is suggested that a wider range of reasoning benchmarks could be used for evaluation.
3. Outdated experimental models: The Deepseek model used in the paper has now evolved to DeepseekV3. Most of the models used in the study date back to before 2024. It is uncertain whether the findings are still applicable to the latest large models.
4. Missing computational efficiency: The paper does not discuss the computational efficiency of the DHAM method (such as training time, inference speed, and parameter quantity). It is unclear from the paper whether the performance improvement is due to an increase in parameters or computational volume.

**Questions:**

1.	Can you provide a theoretical explanation for why early-stage injection produces the smoothest unimodal information flow? Is it because the representations in the early layers are more abstract, or because they exert a stronger "guiding" influence on all subsequent layers?
2.	The stage division is based on representations computed from a specific task dataset (e.g., GSM8K). If a DHAM model trained on GSM8K is directly applied to the MATH task, will its performance degrade? Does this imply that a new stage division is required for each new task, thereby increasing the application cost?
3.	The paper uses saliency based on the gradient of the attention matrix as a proxy metric for measuring information flow. This is a highly valuable perspective, but does this metric only capture the "strength" or "importance" of the information flow, while ignoring its "content" or "semantics"? Could there be a scenario where the saliency curve is smooth, but the model is actually propagating an incorrect reasoning signal?
4.	The paper only cites three references from 2024, with all remaining references dating back to 2023 or earlier. This suggests that the content discussed in the paper is outdated. The models used (such as Deepseek) are also mostly from 2024 or before. Why weren't experiments conducted on newer models? Are there more recent experimental results to support your conclusions? This significantly constrains the novelty of the paper.

---

### Note · Authors · 2025-11-28

I have read and agree with the venue's withdrawal policy on behalf of myself and my co-authors.